# HAPPIER: Hierarchical Polyphonic Music Generative RNN

## Abstract

Generating polyphonic music with coherent global structure is a major challenge for automatic composition algorithms. The primary difficulty arises due to the inefficiency of models to recognize underlying patterns beneath music notes across different levels of time scales and remain long-term consistency while composing. Hierarchical architectures can capture and represent learned patterns in different temporal scales and maintain consistency over long time spans, and this corresponds to the hierarchical structure in music. Motivated by this, focusing on leveraging the idea of hierarchical models and improve them to fit the sequence modeling problem, our paper proposes HAPPIER: a novel HierArchical PolyPhonic musIc gEnerative RNN. In HAPPIER, A higher 'measure level' learns correlations across measures and patterns for chord progressions, and a lower 'note level' learns a conditional distribution over the notes to generate within a measure. The two hierarchies operate at different clock rates: the higher one operates on a longer timescale and updates every measure, while the lower one operates on a shorter timescale and updates every unit duration. The two levels communicate with each other, and thus the entire architecture is trained jointly end-to-end by back-propagation. HAPPIER, profited from the strength of the hierarchical structure, generates polyphonic music with long-term dependencies compared to the state-of-the-art methods.

## 1 Introduction

Drawing inspiration from aesthetic intuition and expressing it as music with specific domain knowledge, composing is a marvelous artistic process for musicians. Due to this uniqueness of composing, although not a new idea, automatic composing remains to be a challenging task. Recent advances in machine learning, especially in sequence generative models, have greatly enabled novel insights into algorithmic music generation and music data analysis (Boulanger-Lewandowski et al., 2012; Gu et al., 2015; Chung et al., 2015; Chu et al., 2016; Engel et al., 2017; Hadjeres et al., 2017; Teng et al., 2017; Jaques et al., 2017; Yu & Varshney, 2017; Thickstun et al., 2017; Roberts et al., 2018).

However, recognizing underlying patterns beneath music across different levels of temporal abstraction while training, and remaining long-term consistency while composing, is still a fundamental challenge for automatic composition algorithms. Thus, most approaches suffer from the lack of coherent global structure of music generated: it lacks consistent theme or structure, and appears to be random and wandering (Oord et al., 2016) (p. 8).

Designing hierarchical architectures to operate on different spatial and temporal resolution scales to ameliorate these problems in machine learning algorithms is not a new idea, and has achieved great success in various fields, including but not limited in the field of computer vision (Lazebnik et al., 2006), reinforcement learning (Dayan & Hinton, 1993; Kulkarni et al., 2016), and in the field of sequence modeling (Hihi & Bengio, 1995; Koutnik et al., 2014; Serban et al., 2016; Mehri et al., 2017).

We note that these kinds of hierarchical architectures correspond with the hierarchical nature of music: the composing of movements and phrases, the progression of chords, and the organization of notes within a measure lie on different temporal resolution levels, from the lowest to the highest. Although sharing some common traits, the patterns of them differ from each other significantly. We present an analogy in writing articles here as an illustration: when writing an article, a mature

writer begins from organizing key ideas of each paragraph given the central idea of the whole article, then he arranges the progression of sentences within a paragraph conditioning on the main idea of the paragraph, and finally writes down words in each sentence given the idea of the sentence with respect to grammar rules. Although the whole article is a unified entity, the patterns and rules within each level, however, vary from each other. Thus, an efficient learning algorithm should be aware of this difference and is supposed to utilize it.

Motivated by this intuition, we propose HAPPIER: a novel HierArchical PolyPhonic musIc gEnerative RNN model to generate music sequentially. For simplicity, we model music as two hierarchies: the higher level measure hierarchy and the lower level note hierarchy. We also assume that every measure in music is specified with a corresponding chord while composing. Under these assumptions, HAPPIER contains a higher-level LSTM (Hochreiter & Schmidhuber, 1997), a variant of RNN, learning correlations across measures and patterns for chord progressions, and a lower-level LSTM learning a conditional distribution over notes. The two hierarchies operate at different clock rates: the higher one updates every measure, while the lower one updates every unit duration. The higher level LSTM gives guidance over the lower level by projecting conditioning vectors to the lower one, meanwhile the lower level LSTM summarizes its cell states to the higher level LSTM once a measure in order to keep the latter one informed. The entire architecture is trained jointly end-to-end by back-propagation.

HAPPIER, gained strength from hierarchical architectures, generates polyphonic music which maintains long-term dependencies, and performs better in listening tests compared to the state-of-the-art methods. Music samples generated by HAPPIER are provided in Appendix C.

## 2 RELATED WORKS

Designing automatic music generation algorithms always attracts great attention from researchers, even dating back to the 80s last century. Attempts include to compose with handcrafted constraints (Ebcioglu, 1988) and to design machine learning algorithms, neural networks for instance (Todd, 1989). However, approaches merely with handcrafted rules generally have unsatisfactory performance because of the lack of variety in the music they generate. Thus, we adopt the learning approach in this work. Analyzing music data also gains great attention in the machine learning community especially recently (Yu & Varshney, 2017; Thickstun et al., 2017; Roberts et al., 2018). These works contribute much either in the perspective of datasets or learning hidden representations for music.

There are two approaches to generate music with respect to the type of data representation: one approach generating raw audio (Oord et al., 2016; Engel et al., 2017), and the other composing music notes, e.g. in the form of MIDI or piano-roll (Boulanger-Lewandowski et al., 2012; Hadjeres et al., 2017; Jaques et al., 2017). Our work belongs to the latter one. In this approach, most works consider monophonic composing (Jaques et al., 2017), but advances have also been made recently in the more challenging polyphonic composing task (Chu et al., 2016). In the light of data representation, our work is similar to the work of Yang et al. (2017), both including simplified polyphonic MIDI representations of a chord track and a melody track.

Recent advances in deep learning have enabled great progress in automatic music generation. Different network architectures and training algorithms have been designed, including Convolutional Neural Networks (Oord et al., 2016; Engel et al., 2017; Yang et al., 2017), Recurrent Neural Networks (RNN) (Boulanger-Lewandowski et al., 2012; Chu et al., 2016; Hadjeres et al., 2017; Jaques et al., 2017; Liang et al., 2017; Lim et al., 2017; Mehri et al., 2017; Teng et al., 2017; Ycart & Benetos, 2017), and Generative Adversarial Nets (Goodfellow et al., 2014) (Yang et al., 2017). However, as most of these works have pointed out themselves, generating music with coherent global structure still remains to be a major challenge for automatic composition algorithms. This difficulty may be primarily attributed to the inefficiency of models to learn long-term consistency in music, which is actually one of the fundamental challenges for most of sequence modeling tasks.

Some of these works have made attempts to tackle this problem. Dating back to the 90s last century, the LSTM (Hochreiter & Schmidhuber, 1997) architecture was proposed to encourage long-term memory in recurrent networks by presenting shortcut connections with gating functions. However, the melodies it generates still remain somewhat random (Jaques et al., 2017) (p. 1). Oord et al.

(2016) designed dilated causal convolution architectures to enlarge the receptive fields of neurons in CNN to learn long-range consistencies. However, the generated samples still vary second-to-second (p. 8). Jaques et al. (2017) regarded the generating process as a sequential decision making process, and adopted a reinforcement learning approach, combining data prior with domain specific knowledge as reinforcement learning rewards to enforce coherent global structures. However, this approach requires handcrafted knowledge of musical theories into the system, which may involve heavy efforts. Hadjeres et al. (2017) gained strength from pseudo-Gibbs sampling to iteratively tune generated chorales. However, since iterative approaches are much more slower than sequential approaches, this method has a rather low generating efficiency.

This paper adopt a hierarchical approach to ameliorate the problem. Our work is related to SampleRNN (Mehri et al., 2017), although their work does not consider automatic music composition problem. Similar to their approach, we use LSTM and different parts of our model run at different clock rates. Unlike their work, the higher hierarchy of our model operates on measures and chords, instead of on frames (p. 2), which are mere concatenations of inputs of the lower hierarchy. We remark that with the higher hierarchy operating on measures and chords, our model can explicitly utilize the knowledge that there are significant differences between the patterns of chord progressions and that of note organizations. Besides, if the number of hierarchies increases, the approach of simply concatenating the inputs of the lower hierarchies as the inputs of the higher ones may not be tractable, instead, some kind of dimension reduction is expected. In this way, we can regard our input of chords as a summarizing of the notes within each measure. Our approach also differs from SampleRNN in that summarizing paths from the lower hierarchy to the higher one is included. The benefits of these paths will be discussed and measured in Section 3 and Section 4.

Chu et al. (2016) also design hierarchical LSTM architectures for composing. However, they first generate melodies, and then accompany the melody with chords and percussion. Instead, we recommend inversely, which is delighted by the remark of Teng et al. (2017) (p. 2) that 'This mimics how classical western Roman-numeral harmony is taught to beginners: only after one has the underlying chord sequence, can one explain the melody in terms of chord tones, passing tones, appoggiaturas, and so on.' Besides, although hierarchical, their approach does not have different parts of the model operating on various temporal resolution scales, and thus still cannot efficiently model the structures of music at very different scales, which finally leads to the lack of hierarchical temporal structure in the music they generate (Teng et al., 2017) (p. 1). While although Teng et al. (2017) first generate chords, and then generate melody based on chords, their model is not end-to-end, and the subparts of their model also do not operate on different clock rates.

## 3 METHOD

In this paper, we propose a hierarchical LSTM (Hochreiter & Schmidhuber, 1997) model HAPPIER for polyphonic music generation. For simplicity, we show a two-level hierarchical LSTM as an example of the model. The model can also be easily generalized to multiple hierarchies in the same way as the two-level model is constructed. Here, we represent polyphonic music as a combination of a chord track and a melody track, where every measure in the music is specified with a corresponding chord. An example of the representation is shown in Figure 1.

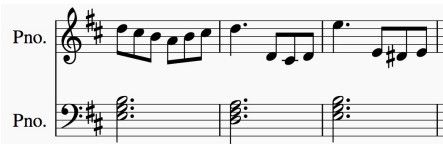

Figure 1: An example of the described representation of polyphonic music. The piece of music is extracted from the **Nottingham** Dataset (Boulanger-Lewandowski et al., 2012).

In previous works focusing on monophonic music generation e.g. (Jaques et al., 2017), composition is generally modeled as a sequential process of generating notes. The process is modeled by a conditional distribution over the next note $n_i$ to generate given all the previous notes generated so far, a deep approximator $f$ parametrized by $\boldsymbol{\theta}$, e.g. an LSTM, is used to approximate this distribution:

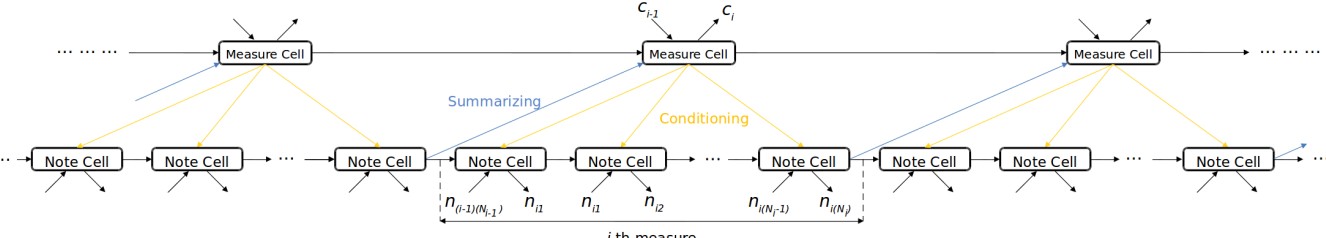

Figure 2: Illustration of the HAPPIER architecture. The loop of the recurrent net is unrolled over time for visualization, and the figure contains 3 measure level loops. A higher-level LSTM learns correlations between measures, and a lower-level LSTM learns a conditional distribution over the notes to generate in each measure given conditioning vectors from the measure level LSTM as guidance. The lower level LSTM summarizes its cell states to the higher one once a measure in order to keep the latter one informed. Different hierarchies operate on different timescales and different temporal resolution levels. The entire architecture is trained jointly end-to-end by back-propagation.

$$p(n_i|n_1, n_2, .., n_{i-1}) \approx f(n_1, n_2, .., n_i; \boldsymbol{\theta}) \tag{1}$$

However, music composition is challenging to these approaches because music contains structures at very different scales, and thus melody generated by approaches with mere LSTM tends to be wandering and random, and lack of coherent global structures (Jaques et al., 2017) (p. 1).

HAPPIER ameliorates the problem by designing end-to-end hierarchical architectures, including a measure level LSTM and a note level LSTM. The two hierarchies operate at different clock rates: the higher one updates every measure, while the lower one updates every unit duration.

Our formulation of the problem is similar to Equation 1, except that we also adopt a hierarchical representation of music. We denote the chord of the $i$-th measure $c_i$, the $j$-th note in the $i$-th measure $n_{ij}$, the number of notes in the $i$-th measure $N_i$, and the total number of measures in a piece of music $N$. The process is defined by jointly modeling Equation 2 via hierarchical LSTM:

$$p(n_{ij}|n_{11}, n_{12}, .., n_{i(j-1)}; c_1, c_2, .., c_{i-1})$$
$$p(c_i|n_{11}, n_{12}, .., n_{(i-1)N_{i-1}}; c_1, c_2, .., c_{i-1}) \tag{2}$$

Figure 2 gives an overview of the architecture. Details of the architecture will be presented in Section 3.1 and Section 3.2.

In correspondence with this hierarchical architecture, we use the following way to represent music data. The way is inspired by the work of Hadjeres et al. (2017), which aligns different tracks well for polyphonic music data representation. In this way, notes are represented by their MIDI pitches and time ticks denoting their beginning and holding; while chords are represented by their pitch encodings and time ticks. Figure 3 below explains this representation method.

One-hot vectors of these representations are used for network inputs. There are 129 types of notes and 79 types of chords in total in the **Nottingham** dataset (Boulanger-Lewandowski et al., 2012). Thus, the prediction problem can be formulated into combinations of classification problems. For simplicity, we still denote notes as $n$ and chords as $c$ in the following sections, rather than separating their pitch and time tick representations. Readers may refer to Appendix B for details.

To train the classifier, the sum of cross entropy losses of chords and notes between the predicted distribution and the actual target distribution is minimized, as shown below:

$$\min_{\boldsymbol{\theta}} \sum_{i \in \{1,2,..,N\}} (-c_i \log(\hat{p}_{c_i}|\boldsymbol{\theta}) + \sum_{j \in \{1,2,..,N_i\}} (-n_{ij} \log(\hat{p}_{n_{ij}}|\boldsymbol{\theta}))) \tag{3}$$

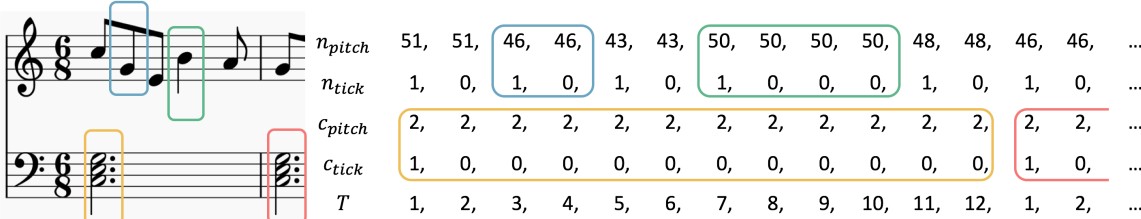

Figure 3: Illustration of the data representation. Boxes in the same color indicate the tokens and their corresponding encodings in the data representation. The piece of music is extracted from the **Nottingham** Dataset (Boulanger-Lewandowski et al., 2012).

While during the generating process, we sample from the predicted distribution to get the next token given all the tokens generated so far. The process is repeated iteratively to generate complete music.

## 3.1 NOTE LEVEL LSTM

The note level LSTM (Note Cells in Figure 2) learns patterns for note generation in the melody track. It operates at a shorter timescale and a higher temporal resolution level. For each unit duration $t_{ij}$, its hidden states $H_{ij}$ and cell states $C_{ij}$ are updated, given the previous note $n_{i(j-1)}$ and previous state vectors $H_{i(j-1)}$ and $C_{i(j-1)}$, together with a conditioning vector $cond_i$ from the measure level LSTM as a guidance for maintaining long-term consistency:

$$H_{ij}, C_{ij} = \mathcal{LSTM}([H_{i(j-1)}, C_{i(j-1)}, n_{i(j-1)}, cond_i]), j \neq 1$$
$$H_{ij}, C_{ij} = \mathcal{LSTM}([H_{(i-1)(N_{i-1})}, C_{(i-1)(N_{i-1})}, n_{(i-1)(N_{i-1})}, cond_i]), j = 1 \qquad (4)$$

A conditional distribution of notes $\mathcal{D}(n_{ij})$ is predicted based upon $H_{ij}$:

$$\mathcal{D}(n_{ij}) = Softmax(Linear(H_{ij})) \qquad (5)$$

At the end of each measure $i$, the note level LSTM projects a summarizing vector $sum_i$ to the measure level LSTM based upon its current cell state $C_{iN_i}$:

$$sum_i = Sigmoid(Linear(C_{iN_i})) \qquad (6)$$

## 3.2 MEASURE LEVEL LSTM

The measure level LSTM (Measure Cells in Figure 2) learns long-term correlations between measures and also learns patterns of chord progressions. It operates at a longer timescale and a lower temporal resolution level. For each measure $t_i$, its hidden states $H_i$ and cell states $C_i$ are updated, given the previous chord $c_i$ and its previous state vectors $H_{i-1}$ and $C_{i-1}$, together with a summarizing vector $sum_{i-1}$ from the note level LSTM to keep it informed:

$$H_i, C_i = \mathcal{LSTM}([H_{i-1}, C_{i-1}, c_{i-1}, sum_{i-1}]), i \neq 1 \qquad (7)$$

A conditional distribution of chords $\mathcal{D}(c_i)$ is predicted from $H_i$:

$$\mathcal{D}(c_i) = Softmax(Linear(H_i)) \qquad (8)$$

A conditioning vector $cond_i$ for the note level LSTM is generated from its current hidden state $H_i$, which is identical within a measure $i$:

$$cond_i = Sigmoid(Linear(H_i)) \qquad (9)$$

### 3.3 HAPPIER: INTEGRATING HIERARCHIES

Intuitively, the two hierarchies can operate on different levels, and can thus make it possible to recognize underlying patterns beneath music across different levels of temporal abstraction. Besides, since the measure level LSTM does not update frequently given its clock rate, it retains long-term correlations, and can thus generate chords with long-term dependencies and guide the note level LSTM to remain consistency over a long range of time steps by projecting conditioning vectors to it.

The two hierarchies interact mutually via the conditioning paths and the summarizing paths. This is also one of the major differences between HAPPIER and SampleRNN (Mehri et al., 2017).

The measure level LSTM guides the note level LSTM to remain long-term consistency by projecting conditioning vectors to it, meanwhile the summarizing paths keep the measure level LSTM informed of events happened downward in the note level LSTM.

The summarizing paths and the conditioning paths as an entity can also be seen as shortcut connections for the note level LSTM. It is known that deep architectures with long forwarding paths involving too many multiplication operations generally suffer from the risk of gradient vanishing or explosion during back-propagation (Pascanu et al., 2012). While shortcut connections can encourage proper back-propagation supervision signals by skipping these multiplication operations, which is the main reason for the outperforming of LSTM (Hochreiter & Schmidhuber, 1997) over conventional RNNs and ResNet (He et al., 2016) over non-residual deep CNNs.

In an LSTM architecture, the gates decide whether to let all the information pass through, when a gated shortcut is 'closed', the functions in the network represent non-short-cut functions, while when a gate is 'open', all the information just passes through, and gradients can thus be back-propagated constantly to keep correlations exist between neighboring tokens as well as between ones apart distantly.

However, the decision of whether to 'open' these gates is learned in an LSTM architecture, and tokens hundreds of time steps away generally lose their correlations because of too many times of 'closing' of these gates. On the contrary, HAPPIER always supports short-cut connections. Consider two note cells hundreds of unit durations away, despite their distance in the note level, it takes only several measures to correlate the two cells in the measure level LSTM: information first goes upward via the summarizing path, then after several time-steps in the measure level LSTM, which has slower clock rates, it can get downward back to the note level LSTM via the conditioning path. Thus, the cooperation of the summarizing paths and the conditioning paths will encourage HAPPIER to learn coherent global structures in music.

Note that SampleRNN (Mehri et al., 2017) does not support these kinds of skip connections, because it does not support information from the lower level LSTM to get upward. Also, without summarizing paths we proposed, the measure level LSTM is modeling $p(c_i|c_1, c_2, .., c_{i-1})$ instead of Equation 2, which means that chords are generated regardless of previous notes generated.

## 4 EXPERIMENTS

Quantitative and qualitative results are shown in this section to measure HAPPIER's sequence modeling performance. We process around 1000 polyphonic tunes into our data representation format shown in Figure 3 from the publicly available **Nottingham** dataset (Boulanger-Lewandowski et al., 2012) for training and validation. Preprocessing and learning details are presented in Appendix B.

### 4.1 QUANTITATIVE ANALYSIS FOR PREDICTION

We first evaluate HAPPIER's sequence modeling capability in a prediction setting. We measure HAPPIER's prediction Negative Log Likelihood (NLL) of the next token given all previous tokens. This does not sufficiently reflect whether HAPPIER can learn long-term consistency over music, but still serves as a quantitative comparison experiment to show how each part of it works.

In this setting, we construct three models for comparison experiments: an LSTM Baseline (only the note level LSTM), a HAPPIER model with only conditioning path, and another HAPPIER model

with both conditioning and summarizing paths. The three models share exactly the same set of architecture details and hyper-parameters. The prediction NLL of the pitch and tick (Figure 3) output by the note level LSTM of the three models are shown in Table 1. Chord NLL is not compared, for the baseline LSTM without hierarchy cannot handle the hierarchical data structure of both a chord track and a melody track. The result indicates that HAPPIER performs better for melody track generation than the LSTM Baseline in the prediction setting, and that the improvement can be mainly attributed to the prescence of the conditioning path.

Table 1: Prediction NLL (Validation)

| Model | Note Pitch NLL | Note Tick NLL |
|---|---|---|
| LSTM Baseline | 0.854 | 0.216 |
| HAPPIER: Only Conditioning | **0.830** | **0.212** |
| HAPPIER: Conditioning + Summarizing | 0.843 | 0.214 |

## 4.2 QUALITATIVE ANALYSIS FOR GENERATION

This Subsection gives qualitative analysis of the generated samples from HAPPIER to show that it generates music retaining long-term consistency. Generated samples are presented in Appendix C, Figure 4 serves as an example of them.

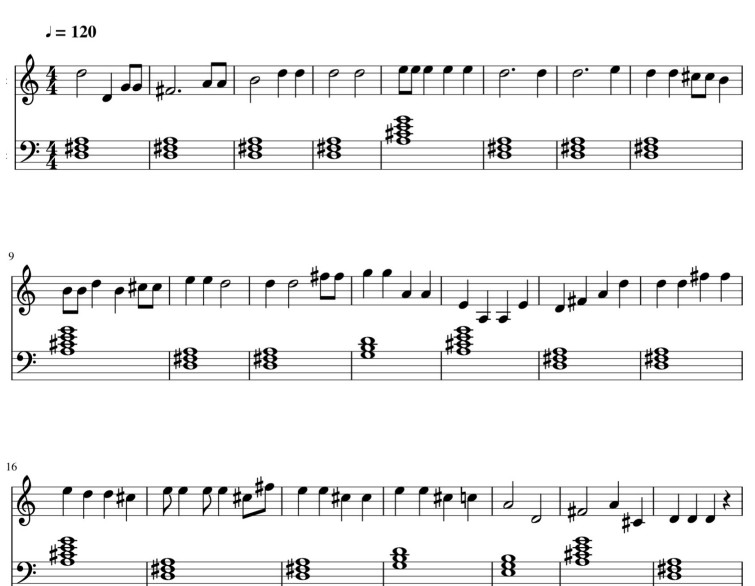

Figure 4: One extracted generated sample by HAPPIER. The full sample and some more samples are presented in Appendix C.

In Figure 4, the generated tune remains in D major (e.g. given the #C and #F) globally. The progression of chords also follows D major. The melody track returns to the tonic note of D every a while and the chord track returns to the tonic chord correspondingly. Generated samples from HAPPIER retain long-term consistency in this way, and we also suggest our readers to listen to the generated samples themselves with comparison to the state-of-the-art methods to judge this [1].

---

[1]Links to the generated samples of the state-of-the-art methods are presented in Appendix A.

### 4.3 LISTENING TESTS

We also evaluate the performance of HAPPIER by human listening tests. We compare generated samples by HAPPIER with those from RNN-RBM (Boulanger-Lewandowski et al., 2012), Deep-Bach (Hadjeres et al., 2017) and SequenceTutor (Jaques et al., 2017). These three works all publish their generated samples online (Appendix A). These publicly accessible samples are used for this experiment. RNN-RBM is trained on similar datasets as ours, while DeepBach and Sequence Tutor are trained on different datasets. Sequence Tutor is trained with more than 30000 MIDI files, which are not made publicly accessible, and DeepBach is trained with 352 Bach Chorales.

We conduct AB preference tests among the generated samples of HAPPIER and those of the 3 models mentioned above. We also conduct a test between our generated samples and samples from our training dataset. Each subject is shown a set of 4 pairs of samples, and each pair contains samples from two different models. Subjects then choose the preferred samples or choose not to prefer to any of them (no pref.) after listening. Our samples are rendered with GarageBand from the MIDI files the network generates.

Subjects are also asked to give information about their musical expertise. They could choose a category fits them best among Level 1, knowing little about music theories; Level 2, music lovers or instrument players; and Level 3, having received training in composition. For this experiment, 179 subjects take the test, among whom 103 in Level 1, 64 in Level 2, and 12 in Level 3. Experiment results are shown in Figure 5. Results show that HAPPIER is more auditorily pleasing compared with the state-of-the-art approaches in machine learning community, and the generated samples from HAPPIER can be hardly distinguished from samples from the **Nottingham** dataset.

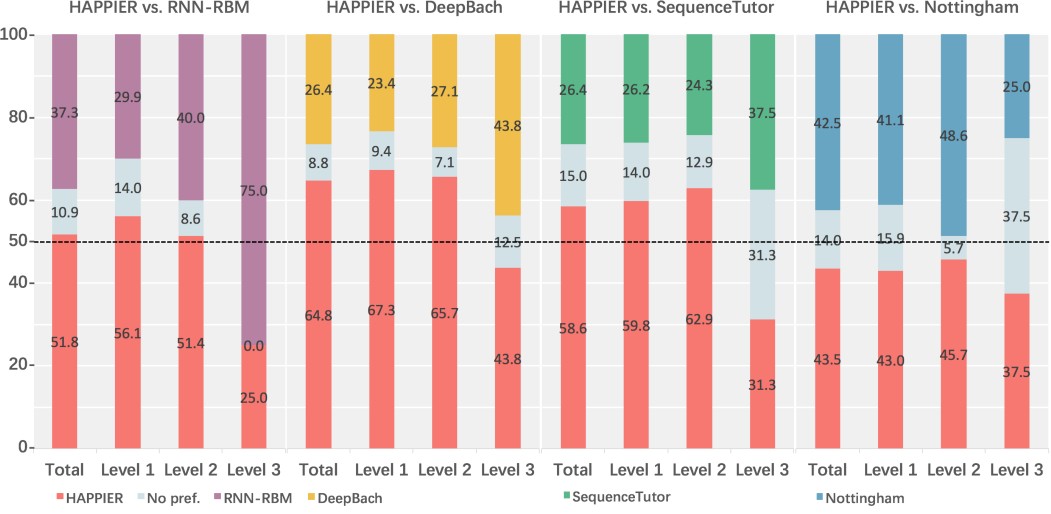

Figure 5: Experiment results for the listening test. 179 subjects are involved.

## 5 CONCLUSION

We have proposed a novel end-to-end hierarchical polyphonic music generative RNN: HAPPIER, which gains strength from the correspondence of its hierarchical structure and the hierarchical nature of music. The model learns long-term correlations and performs better in listening tests compared to the state-of-the-art methods. The work mainly focuses on the automatic composition problem, but our contribution is not limited there. We believe the HAPPIER approach of designing hierarchical deep recurrent models with sub-parts of it operating on different clock rates and on different temporal resolution levels will be promising for a number of sequence modeling applications, where conventional approaches generally suffer from their inefficiency of learning long-range correlations.

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

## APPENDICES

### APPENDIX A: LINKS TO GENERATED SAMPLES BY THE STATE-OF-THE-ART MODELS

RNN-RBM and Nottingham Dataset (Boulanger-Lewandowski et al., 2012) :

```
http://www-etud.iro.umontreal.ca/~boulanni/icml2012;
```

DeepBach (Hadjeres et al., 2017) :

```
https://sites.google.com/site/deepbachexamples;
```

Sequence Tutor (Jaques et al., 2017) :

```
goo.gl/XIYt9m;
```

APPENDIX B: IMPLEMENTATION DETAILS

We process the **Nottingham** dataset (Boulanger-Lewandowski et al., 2012), a collection of around 1200 folk tunes with chords, instantiated from MIDI format. The tunes have an average number of measures of 37. Most of them are composed of two tracks: a melody track and a chord track, as shown in Figure 1. Tunes not fit into this format and tunes with rare complex time signature, 9/8 for instance, are discarded for simplicity. Finally, 993 tunes remain as our training, validation, and test datasets.

We discretize time with 16th notes or 32nd notes: each time tick in our model equals to the duration of an 16th note or an 32nd note, which is called an unit duration. In this way, for example, each measure is subdivided into 12 time ticks, for time signatures 6/8 and 3/4; or 16 time ticks, for 2/4 and 4/4.

The input of the note level LSTM $n$ contains concatenated one-hot series of $n_{pitch}$, $n_{tick}$, and $T$ (Figure 3), and the input of the measure level $c$ is one-hot series of $c_{pitch}$.

While training, the time ticks every measure is fixed as a constant 12 or 16 to simplify the training process. To implement this, two identical networks sharing the same set of parameters are constructed, note cells of the first one update 12 times within the inner loop of a measure; while the other 16 times (Figure 2), each fed with corresponding data. In the training process, teacher forcing (Williams & Zipser, 1989) technique is adopted.

In Equation 3, we simplify the representation by integrating pitch and time-tick losses of the note level cells, a more exact loss function is presented below:

$$
\min_{\boldsymbol{\theta}} \sum_{i\in\{1,2,..,N\}} (-c_{pitch_i}\log(\hat{p}_{c_{pitch_i}}|\boldsymbol{\theta})+
$$
$$
\sum_{j\in\{1,2,..,N_i\}} (-n_{pitch_{ij}}\log(\hat{p}_{n_{pitch_{ij}}}|\boldsymbol{\theta}) - n_{tick_{ij}}\log(\hat{p}_{n_{tick_{ij}}}|\boldsymbol{\theta}))) \tag{10}
$$

While generating, note that the sampled token of $n_{pitch}$ and $n_{tick}$ may not correspond. For instance, consider the case that the generated $n_{pitch}$ has changed from the previous pitch but $n_{tick}$ is 0, denoting that the previous pitch should hold. When this happens, the generated $n_{tick}$ is adopted and the generated $n_{pitch}$ is thus replaced by the previous pitch to correspond with the holding.

Both the measure level and the note level LSTM have hidden state sizes of 60. The sizes of both the conditioning vector and the summarizing vector are set to 10. Optimization is performed with Adagrad (Duchi et al., 2011), a batch size of 16, initial learning rate of 0.7, and a stepwise learning rate decay of 0.9 every 300 steps. Gradients are clipped to ensure the L2 norm was less than 5.0, and weight regularization is applied with $\beta = 2.5 \times 10^{-5}$. The weights are initiated from a Gaussian distribution $\mathcal{N}(0, 3 \times 10^{-2})$. The losses for the first 4 tokens of each sequence are not used to train the model, since it cannot reasonably be expected to accurately predict them with no context (Jaques et al., 2017). Maximum back-propagation truncated length is set to be 12 measures.

We implement HAPPIER with TensorFlow, music21 (Cuthbert et al., 2010) and MuseScore libraries.

APPENDIX C: GENERATED SAMPLES

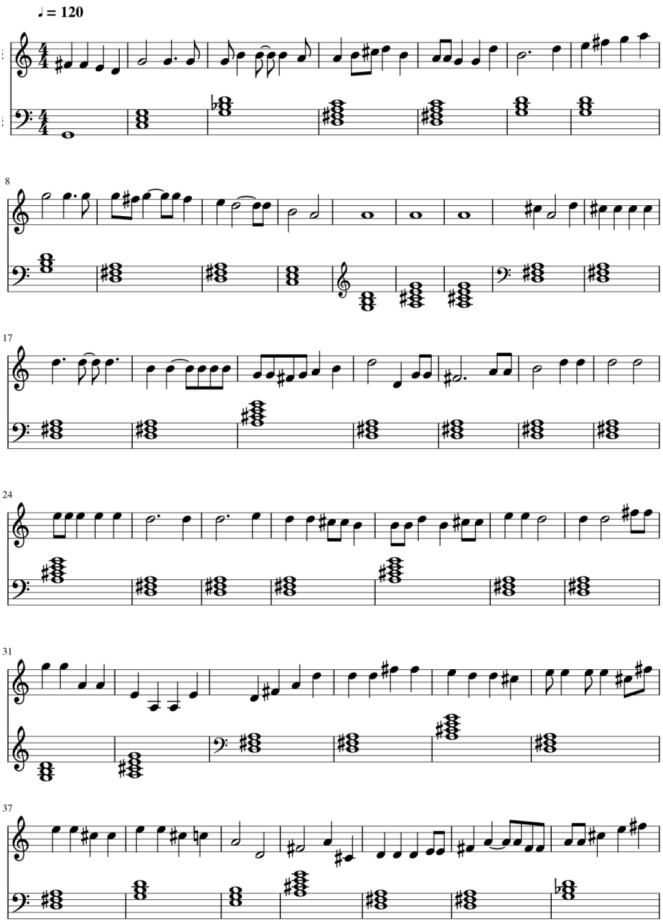

Figure 6: Generated sample.

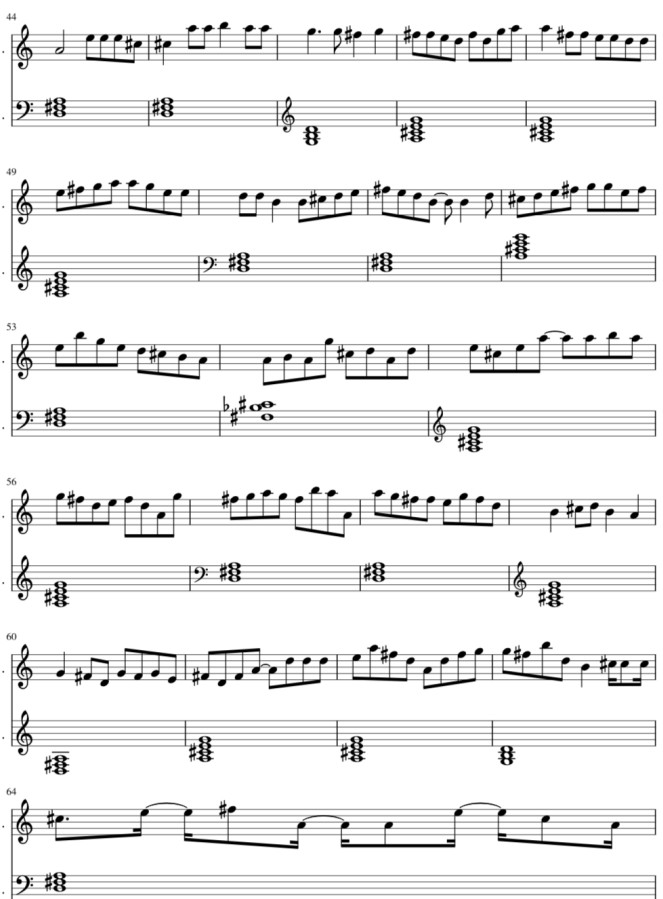

Figure 7: Generated sample.

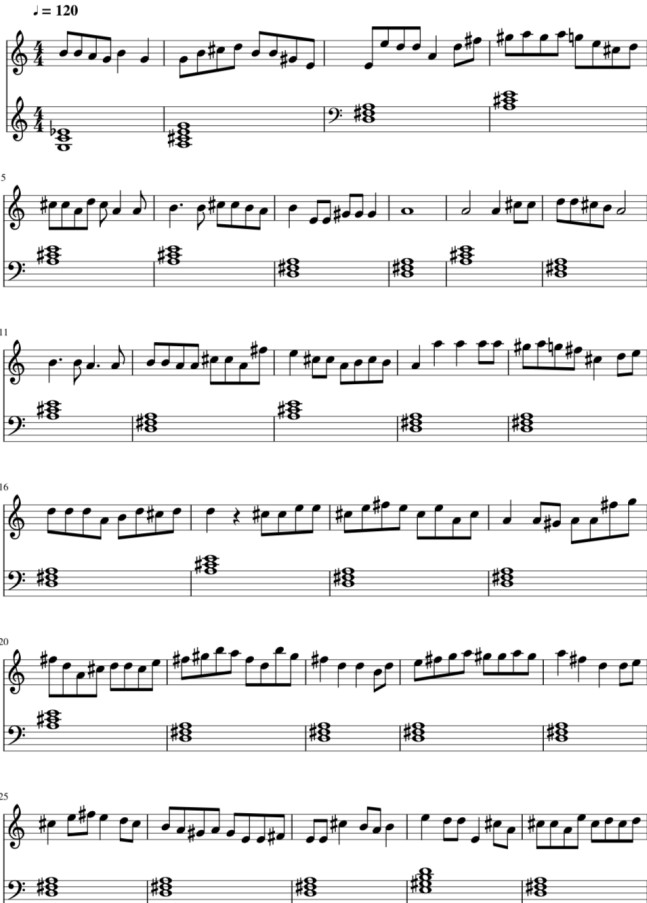

Figure 8: Generated sample.

