# OpenReview forum: "HAPPIER: Hierarchical Polyphonic Music Generative RNN"
_ICLR.cc/2019/Conference_

### Official Review · AnonReviewer2 · 2018-11-01
**Straight forward idea with poor evaluation**

**Rating:** 3
**Confidence:** 5

**Review:**

The authors propose a hierarchical model of symbolic music that takes explicit advantage of measures and chords to construct the hierarchy. Their model is very similar to SampleRNN (2-level RNN Autoregressive Model) but with an additional cross-entropy loss for chord labels at the higher level and a summarization connection passing back to the high level from the low-level at the end of each bar. They show that given monophonic music with chord labels their model is able to produce reasonably coherent chords and note samples, and improves the NLL over a low-level model alone.

The core of their approach (using measures as a natural hierarchy for a multi-level RNN) is a good one, but not new in of itself as it was the basis for the prior work of Roberts et al. (http://proceedings.mlr.press/v80/roberts18a/roberts18a.pdf). The authors highlight in section 3.3 that their work is distinguished by the summarization connection, but do not provide any evidence in their results that the connection is useful. They find in Table 1 that connection hurts NLL on the note level, and do not compare summarized to non-summarized models in the listening tests.

The area for most improvement in the paper is the evaluation, especially the listening tests. The authors compare samples from four models that generate different types of outputs and were trained on different datasets. Because of this, the notion of user preference is completely convoluted with external factors. In particular the comparisons to DeepBach and SequenceTutor are inappropriate and give little information about the quality of the model architecture itself. To be useful comparisons should be restricted to model architectures that are trained on the exact same data as HAPPIER, and output both chords and melodies like HAPPIER does. Given that the novelty of the paper rests on the summarization connections, and they were not shown to help NLL, it would be natural to try and compare the different model variants in the paper and see if the NLL misses some element of larger structure that listeners may care about. My rating is thus based on the lack of novelty and poor quality of evaluation justifying the actual novel aspects of the paper.

Some minor comments that could also help improve the paper:

* Including NLL for chords is important to compare summarization (does it help in chord prediction?)
* The input representation could use further clarifying. What is the dictionary of chords to predict from? Are they just chord names or individual notes (the figures imply notes, but that doesn't seem what's happening). In Figure 2, clarify the meaning of tick, what 1, 0 means in terms of time progression.
* Provide quantitative evidence for the claims in 4.2 that the notes and chords belong to the same key. Compare real data and generated data for those statistics.
* Provide explanation for why Note NLL is higher for Summarization.
* Minor notation problems: Eq 1, f should not be a function of n_i. Similar, in Eq 2, p(n_{ij}) should be a function of c_i. Eq 3 doesn't define what the hat represents.

---

### Official Review · AnonReviewer3 · 2018-11-02
**approach is OK, but needs (better) results**

**Rating:** 3
**Confidence:** 4

**Review:**

PRO's:
+good problem: generating polyphonic music with long-term structure
+reasonable approach: modification of SampleRNN: makes sense

CON's:
-doesn't work.

My fundamental critique of this paper is that, while the authors claim that their system " generates polyphonic music which maintains long-term dependencies", in fact what it generates it is not really polyphonic, nor-- more importantly-- does it demonstrate the kind of long-term structure present in the training set.

1) Polyphony: The model predicts a combination of monophonic melody plus chords (i.e. chord names such as "A+", "C7" etc). This is different from polyphony, in which the model would predict the actual voicing used for those chords. However, this could be seen as an error in terminology; if the authors claimed that they were predicting a monophonic musical voice plus chords, and they did that well, that would be absolutely fine. Generating a coherent melodic line that continues, along with the chords underneath it, would be a great achievement. However, that is not what happens here. For examples, in the provided examples, e.g. Measure 19 of Fig 4, Measures 1,3,4,5, ... of Fig 6, contain stylistically unusual combinations of chords and melodic lines. By "stylistically unusual", I simply mean that those examples are not consistent with the Nottingham dataset.  Furthermore, in my subjective opinion, the examples that I listed above also just don't really musically work. There is no question that in the right context, any of those particular combinations of chords and melody notes *could* be made to work: for example, the first measure of Fig 6 would be perfectly fine as the beginning of a different song. (E.g. it could be taken as a slight reharmonization of the opening of "lullaby of birdland", but that would require a coherent continuation. )

2) Long-term structure: It seems to me that one of the key things that this paper sets out to do is to get strong long-term dependencies. The motivation for the SampleRNN-inspired approach is to have generation at multiple time scales, for example. However, there is no evidence in the presented examples of long-term structure. Consider Fig 6, for example. Where is the long-term structure? A D major chord is frequently repeated with occasional A7. That is reasonable but it does not necessarily demonstrate long-term structure, anymore than learning that "q" is often followed by "u" demonstrates long-term structure. There is no melodic motif, there is no sense of 4-bar phrasing (or any other recurring such pattern that I can tell). In fact, all of the samples shown (Fig 4, 6, 7, 8) all end up with the chord D major played most of the time, after what appears to be a bit more variation in the first few chords.

The results of the listening test are strange to me (beyond some of the apples-to-oranges comparisons). I cannot comment on those without hearing the pairs of examples that were actually played. How were those pairs selected?

At the moment, it does not seem worthwhile for this review to get into details about exactly how the system works, in light of the problematic output. If there is reason for me to do so, I would gladly oblige. The authors do make a variety of choices that appear to be fairly sensible.

I would very much look forward to seeing a revised version of the system in future that produces the  kind of output that the system is intended to produce (and described as producing).

---

### Official Review · AnonReviewer1 · 2018-11-02
**Interesting but not novel**

**Rating:** 2
**Confidence:** 4

**Review:**

This paper proposes a hierarchical RNN, where the first layer is note-level and the second level is measure-level. In an experiment on the Nottingham MIDI dataset, they show slight improvements in log-likelihood.

Overall:

This is an interesting application of hierarchical RNNs. However, hierarchical RNNs are known to improve performance. This is an application of existing work (for example, Alexander Graves' thesis also uses hierarchical RNNs and shows improved performance). For an applications-oriented paper, I would hope to see many more experiments than just one on a tiny dataset, and improvements in log-likelihood that are more than the marginal improvements reported here. The human evaluation is neat but is inconclusive–in a glaring act of omission, the authors do not link to samples generated by their model, while they include samples generated by the competition. For a fair review, one would hope to compare the models side by side to qualitatively judge the reliability of the MTurk experiments.

Minor nits:

I appreciate the human evaluation experiments on MTurk but they are very difficult to understand with the figure 5. Please label the y-axis. Think of a different way to present the results. Do not include the numbers on the bars.

The acronym HierArchical PolyPhonic musIc gEnerative RNN is destructive; it devalues useful acronyms. Please do not use it.

The paper has many grammatical and spelling errors. Please hyphenate compound adjectives.

---

### Public Comment · ~Bob_L._Sturm1 · 2018-10-04
**Sensible idea but the conclusions are not supported by the evidence**

I like what the authors are trying to do here. Taking a hierarchical approach to modeling music makes sense. This is in line with the multiple viewpoint approach (e.g., seehttps://www.tandfonline.com/doi/abs/10.1080/09298219508570672). But the conclusions made in the paper are not supported by the presented evidence. The generated examples do not show any success of modeling the dataset used. More detailed comments here: https://highnoongmt.wordpress.com/2018/10/02/going-to-use-the-nottingham-music-database/

---

### Meta-Review · Area_Chair1 · 2018-12-14
**rejection**

**Confidence:** 5
**Recommendation:** Reject

**Metareview:**

Although all the reviewers find the problem and the approach of using hierarchical models important and interesting, how it has been executed in this submission has not been found favourable by the reviewers.